# Evolution of the Jawed Vertebrate (Gnathostomata) Stomach Through Gene Repertoire Loss: Findings from Agastric Species

**DOI:** 10.3390/jdb13030027

**Published:** 2025-08-05

**Authors:** Jackson Dann, Frank Grützner

**Affiliations:** 1School of Biological Sciences, The University of Adelaide, Adelaide, SA 5005, Australia; jackson.dann@adelaide.edu.au; 2Robinson Research Institute, The University of Adelaide, Adelaide, SA 5005, Australia

**Keywords:** molecular evolution, gene loss, vertebrates, gastric development, comparative genomics

## Abstract

The stomach has been a highly conserved organ throughout vertebrate evolution; however, there are now over 20 lineages composed of monotremes, lungfish and teleost fish displaying a secondary loss of stomach function and morphology. This “agastric phenotype” has evolved convergently and is typified by a loss of gastric glands and gastric acid secretion and a near-to-complete loss of storage capacity of the stomach. All agastric species have lost the genes for gastric enzymes (*Pga* and *Pgc*) and proton pump subunits (*Atp4a* and *Atp4b*), and gastrin (*Gast*) has been lost in monotremes. As a key gastric hormone, the conservation of gastrin has not yet been investigated in the lungfish or agastric teleosts, and it is unclear how the loss of gastrin affects the evolution and selection of the native receptor (*Cckbr*), gastrin-releasing peptide (*Grp*) and gastrin-releasing peptide receptor (*Grpr*) in vertebrates. Furthermore, there are still many genes implicated in gastric development and function which have yet to be associated with the agastric phenotype. We analysed the evolution, selection and conservation of the gastrin pathway and a novel gastric gene repertoire (*Gkn1*, *Gkn2*, *Tff1*, *Tff2*, *Vsig1* and *Anxa10*) to determine the correlation with the agastric phenotype. We found that the loss of gastrin or its associated genes does not correlate with the agastric phenotype, and their conservation is due to multiple pleiotropic roles throughout vertebrate evolution. We found a loss of the gastric gene repertoire in the agastric phenotype, except in the echidna, which retained several genes (*Gkn1*, *Tff2* and *Vsig1*). Our findings suggest that the gastrin physiological pathway evolved differently in pleiotropic roles throughout vertebrate evolution and support the convergent evolution of the agastric phenotype through shared independent gene-loss events.

## 1. Introduction

The stomach is a highly conserved and integral organ throughout jawed vertebrate (Gnathostomata) evolution, as evidenced by conserved developmental pathways found in extant representatives of basal Gnathastomes [1] and even dating back to protochordate species [2,3]. Though morphological features can vary widely between taxa, comparable components include storage compartments (enlargement of the lumen) and acid-secreting glandular epithelium [4,5]. Despite gastric features being nearly ubiquitous, a unique gastric phenotype has evolved over 15 times independently in ray-finned fish (Actinopterygii) and at least twice in lobe-finned fish (Sarcopterygii): once in lungfish and once in monotremes. This gastric phenotype—hereby known as agastric—is typified by an absence of acidic luminal contents, loss of glandular epithelium and a significant reduction in overall size and gross morphology [6,7]. These convergent morphologies only derive slightly in monotremes, where the echidna has retained the pyloric sphincter and some storage capacity [8].

In line with convergent phenotypes, agastric genotypes also exhibit similarities: agastric species share the concomitant loss of genes for gastric enzymes (*Pga* and *Pgc*) and proton pump subunits (*Atp4a* and *Atp4b*). Previously, it was found that monotremes also have lost gastrin (*Gast*): a key gastric hormone regulated by the upstream neuropeptide gastrin-releasing peptide (*Grp)* and its native receptor (*Grpr*) [7]. When released from gastric G cells, gastrin binds to the native cholecystokinin B receptor (*Cckbr*) on parietal cells, thereby potentiating gastric acid secretion [9,10,11,12]. Given the heterogeneity of the gastric epithelium in cell type (e.g., parietal cells, mucous cells and stem cells) and function (e.g., acidic digestion, immunity and hormone release) and various physiological and intracellular signalling pathways regulating gastric development and function, we expect many more genes to be correlated with the drastic shift in gastric anatomy and physiology [13,14,15].

Recent discoveries in vertebrate developmental genetics and the availability of high-quality whole genomes have expanded the repertoire of genes implicated in gastric development and function in various species. These have included the constituents of the gastrokine and trefoil factor families (*Gkn1*, *Gkn2*, *Gkn3*, *Tff1* and *Tff2*), and secreted products of gastric mucosal epithelia with both independent and synergistic anti-inflammatory tumour-suppressing and anti-apoptotic roles [16,17]. In addition, *Vsig1*—a junction adhesion molecule expressed highly in gastric epithelia—and annexin A10 (*Anxa10*)—a component of the phospholipid-binding protein annexin superfamily—have displayed restricted gastric epithelial expression and have been implicated in squamous versus glandular epithelial differentiation [18,19,20]. Despite some extensive work on expression patterns (particularly in gastrokine and trefoil factor families) and recent work by Kato et al., (2024) [21]—which noted the absence of *Vsig1* in agastric teleost lineages and the platypus—important questions remain. In particular, it is still unclear what emergent properties (i.e., gene ontologies) arise from the list of lost genes and whether these additional genes are conserved in agastric vertebrate lineages [22,23].

To better understand the evolution of the agastric phenotype in the Actinopterygii and Sarcopterygii clades, we explored the evolution, selection and loss of the gastrin physiological pathway along with the conservation of novel gastric gene sets in the genomes of gastric and agastric taxa.

## 2. Materials and Methods

### 2.1. Sequence Retrieval and Selection of Gastric Genes

A total of 21 chromosome-level genome assemblies from evolutionary representative taxa of tetrapods (placental mammals, marsupials, monotremes and reptiles), Sarcopterygii (lobe-finned fish) and Actinopterygii (ray-finned fish) were downloaded from NCBI Genbank (https://www.ncbi.nlm.nih.gov/genbank/ (accessed on 5 April 2023) with a scaffold N50 filter of above 10 Mb. Aside from gastrin physiological pathway constituents, gastric genes were chosen for analyses by implication in gastric expression, digestion, gastric mucosal maintenance or gastric development from the previous literature on vertebrates in combination with manual searching through the gene ontology resource and KEGG pathway database [7,9,14,24,25]. From this list, genes were selected for gene-loss analyses if NCBI Genbank contained no annotated gene or protein in one or more agastric species.

Annotated coding DNA sequences (CDSs), exon sequences and protein sequences for gastrin pathway constituents (GAST, CCKBR, GRPR and GRP) and other gastric genes (GKN1, GKN2, TFF1, TFF2, VSIG1 and ANXA10) for these species were downloaded from NCBI Genbank (https://www.ncbi.nlm.nih.gov/genbank/). Where annotated, protein sequences were absent, CDSs from genomic sequences were uploaded to Geneious 11.0.14+1 for manual translation and alignment. See Appendix A for sequence and genome accession numbers for all species used.

### 2.2. Gene-Loss Analysis

To establish gene orthology and loss, we took a modified approach of existing bioinformatic pipelines [26,27]. We firstly established genomic neighbourhoods (synteny) for the target gene through the identification of consistent flanking orthologs between species. Where synteny was conserved (i.e., flanking genes were present with no gaps), the region between orthologs was used as a query for a tblastx search with the CDS from a closely related species [28]. Where one or no flanking genes were conserved, the entire genome was used as a query for the tblastx search. To validate these sequences, the top results from BLAST 2.16.0 were then downloaded and used as a query against raw reads in the sequence read archive (SRA) BLAST function [29]. The top result was then imported into Geneious and aligned with the query CDS using Clustal Omega 1.2.2 on default settings. Inactivating mutations (e.g., premature stop codon, frameshift and deletion) were then manually annotated from these alignments. Where no sequence was returned from the BLAST search but synteny was conserved, the sequence was assumed to be entirely eroded.

### 2.3. Sequence Alignments and Phylogenies

Multiple sequence alignments were performed using the Clustal Omega plugin in Geneious 11.0.14+1, except for deleted/pseudogenised monotreme genes [30]. Protein maximum likelihood phylogenies were then constructed using the IQ-TREE web server (http://iqtree.cibiv.univie.ac.at/ (accessed on 12 September 2023)) tree inference tool with standard settings and 1000 bootstrap replicates [31]. Substitution models derived from the Bayesian information criterion and alignment outgroups can be found in Appendix A.

### 2.4. Selection Analysis (Ka/Ks Ratios)

To detect signatures of selection, amino acid alignments and corresponding open reading frame DNA alignments were converted into PAML format codon alignments using PAL2NAL v 14 [32]. Codon alignments along with their corresponding phylogenies in Newick format were then input into EasyCodeML v1.41 using the preset branch model with multiples models of foreground branch selection for each lineage and likelihood ratio tests (LRTs) to assess model significance [33]. EasyCodeML outputs can be found in Appendix A.

## 3. Results

### 3.1. Conservation and Loss of the Gastrin Release Pathway and Novel Gastric Gene Repertoire in Gastric and Agastric Jawed Vertebrates (Gnathostomata)

Synteny analysis of gastrin (*Gast*) and the native gastrin receptor (*Cckbr*), as well as the gastrin-releasing peptide (*Grp*) and its receptor (*Grpr*), returned conserved syntenic blocks in most ray- and lobe-finned fish species with recurring differences of flanking loci between lineages (Figure 1A,B). Our genome search returned no orthologous sequences for gastrin in the genomes of the agastric pufferfish or West African lungfish, indicating near-total erosion in these genomes. In the agastric zebrafish, an orthologous sequence in the conserved syntenic region which could be validated against raw sequence reads was identified. The highly conserved syntenic blocks of tetrapod species corroborated the previously noted gene losses of gastrin in both monotreme species (Ordoñez et al., 2008 [9]), with our analyses indicating the loss of *Grpr* in the platypus through deletion of the latter two exons (Figure 1C).

These findings suggest that the agastric phenotype does not correlate with the loss of gastrin. Furthermore, the absence of gastrin is not consistently associated with the loss of its indirect upstream regulator, *Grpr*.

We then explored the conservation and loss of genes implicated in gastric epithelial maintenance and development in jawed vertebrates. A BLAST search and synteny analysis of the sequences bearing gastrokine family homology (*Gkn1*, *Gkn2* and *Gkn3*), trefoil factor homology (*Tff1* and *Tff2*), as well as *Anxa10* and *Vsig1* returned no orthologous sequences in agastric Actinopterygii taxa. Conversely, in the gastric grey bichir and spotted gar, there was a conservation of the syntenic neighbourhood with an orthologous sequence conserved (Figure 2A).

In Sarcopterygii, gastrokine family paralogs (*Gkn1*, *Gkn2* and *Gkn3*) were mainly conserved together within a syntenic block flanked by *Antxr1* and *Bmp10*. Gastrokine paralogs 2 and 3 were lost in both monotreme species, but *Gkn1* was retained in the echidna. In the platypus, *Gkn1* was found to be inactivated through the accumulation of missense and nonsense mutations throughout the CDS. The three Gastrokine paralogs found in the West African lungfish were not placed within the conserved genomic region, and further phylogenetic analysis of these sequences shows that basal Sarcopterygii sequences (coelacanth, West African lungfish) were placed in a clade separate from all three vertebrate paralogs. Furthermore, the echidna *Gkn1* sequence has higher similarity to *Gkn3* sequences and was placed within the vertebrate *Gkn3* clade with 100 percent bootstrap support (Appendix A).

Trefoil factor paralogs 1 and 2 (*Tff1* and *Tff2*) were absent in the coelacanth and West African lungfish and found only in tetrapod species in a highly conserved syntenic block between *Tff3* and *Spock3*, except in the platypus—which was missing both—and the echidna, which was missing *Tff1*. Genomic sequences of *Anxa10* and *Vsig1* were found in all Sarcopterygii taxa but not in the platypus, supporting the findings by Kato et al., [21], which associate the loss with the agastric phenotype (Figure 2B). These findings suggest that these genes were likely present in early gnathastome evolution, prior to the divergence of major jawed vertebrate lineages, and were later lost independently.

### 3.2. Sequence Evolution and Selection Pressures in the Gastrin Release Pathway Between Gastric and Agastric Jawed Vertebrates (Gnathostomata)

We then investigated sequence evolution and Ka/Ks ratios in the gastrin release pathway in gastric and agastric vertebrates to see whether gastrin loss affected the evolution or selection of other pathway components.

The Actinopterygii clade of gastrin sequences displayed minimal differences in substitution per site between the gastric three-spined stickleback (0.48) and agastric Japanese rice fish (0.34) (Figure 3A). Branches leading to monotreme and therian mammal CCKBR clades displayed identical lengths (0.17 substitutions per site) but differed more in post-divergence in the mouse and wombat (0.41 and 0.25 substitutions per site) when compared to the agastric platypus and echidna (0.07 and 2 × 10^−6^ substitutions per site). Actinopterygii CCKBRA sequences had similar substitutions per site within the clade of the pufferfish (0.15), the Japanese rice fish (0.15) and the three-spined stickleback (0.12). The pufferfish CCKBRB sequence contained a larger number of substitutions per site (0.2) when compared to three-spined stickleback (0.07) and Japanese rice fish (0.13) (Figure 3B).

In the Sarcopterygii clade, *Grp* sequences from the coelacanth and West African lungfish displayed large variations in substitutions per site (0.2 vs. 0.6). Though echidna and platypus sequences displayed identical estimated substitutions per site (0.06), there was a substantial difference in branch length relative to the root of mammalian sequences between monotremes (0.16), marsupials (0.44) and placental mammals (0.7). In the Actinopterygii clade, branch lengths differed relative to the root of the clade with the Japanese rice fish (0.76), pufferfish (0.41) and three-spined stickleback (0.55) (Figure 3C). Sarcopterygii and Actinopterygii *Grpr* sequences exhibited little difference in substitutions per site between gastric and agastric species, as shown by comparisons between the salmon (0.15) and the pufferfish (0.18) and comparisons of the coelacanth (0.08) and West African lungfish (0.07) (Figure 3D). Together, these findings suggest that the presence or absence of a stomach does not strictly correlate with accelerated molecular evolution or pseudogenization of these genes across vertebrate lineages. Instead, some agastric species retain high sequence conservation, implying possible pleiotropic roles or evolutionary constraints unrelated to gastric function.

Finally, we calculated the ratio of synonymous to non-synonymous substitutions (Ka/Ks) using maximum-likelihood branch model selection analyses. Overall, we found that the *Gast, Cckbr*, *Grp* and *Grpr* genes displayed several bouts of purifying selection (Appendix A). Foreground scenarios with significant likelihood-ratio tests (LRTs) in the *Cckbr* phylogeny included branches leading to the clade with Japanese rice fish, pufferfish and three-spined stickleback *Cckbrb* (*p* = 0.04, ω1 = 0.19 and ω0 = 0.08), the three-spined stickleback *Cckbra* (*p* = 0.01, ω1 = 3 × 10^−3^ and ω0 = 0.07) and *Cckbrb* sequences (*p* = 1.12 × 10^−5^, ω1 = 0.33 and ω0 = 0.08). In the *Grp* phylogeny, significant LRTs included branches leading to the tetrapod clade (*p* = 2.17 × 10^−5^, ω1 = 8.9 × 10^−4^ and ω0 = 0.28), the mammalian clade (*p* = 0.03, ω1 = 0.02 and ω0 = 0.24) and the mouse sequence (*p* = 0.04, ω1 = 0.02, and ω0 = 0.24). The *Grpr* phylogeny included two significant LRTs: the branch leading to the pufferfish sequence (*p* = 0.02, ω1 = 0.11 and ω0 = 0.04) and to the Sarcopterygii clade (*p* = 8.93 × 10^−7^, ω1 = 1.94 × 10^−3^ and ω0 = 0.05) (Table 1). These findings suggest that independent bursts of sequence evolution in the gastrin release pathway were primarily characterised by conservative mutations resulting in the maintenance of functional integrity. Furthermore, these molecular changes appeared to have no direct correlation with variation in gastric phenotypes.

## 4. Discussion

Stomach anatomy in vertebrates can be broadly described as gastric or agastric. The latter phenotype lacks gross morphology, glandular epithelium and gastric acid secretion, as has been described in independent teleost lineages and monotremes. Changes in stomach morphology have been correlated with the loss of gastric enzymes (*Pga* and *Pgc*), proton pump subunits *(Atp4a* and *Atp4b*) and the gastrin hormone (*Gast*) in monotremes [7,9,11,21]. Due to the complex constitution and function of the stomach—with a large complement of cell types, signalling pathways and physiological functions—we hypothesised that the gene losses associated with the agastric phenotype extended beyond the aforementioned genes. To test this, we investigated the evolution, selection and loss of the gastrin physiological pathway along with genes implicated in gastric development and functionality in both gastric and agastric Sarcopterygii and Actinopterygii taxa.

### 4.1. Gastrin Physiological Pathway: Correlations with Gastric Phenotype, Known Functions and Genetic Dispensability

We determined loss of gastrin in four of the five agastric species presented (platypus, echidna, pufferfish and West African lungfish) but no substantial differences in substitution rate or selection pressures of the gastrin physiological pathway constituents (*Gast*, *Cckbr*, *Grp* and *Grpr*) between gastric and agastric species or between species who had retained or lost gastrin. These findings demonstrate that gastrin loss is not likely associated with the agastric phenotype and that loss of gastrin does not correlate with subsequent sequence evolution, gene loss or selection in other gastrin release pathway constituents.

These findings are surprising given both the conserved role in gastric acid secretion and localised expression of gastric mucosa in vertebrate species [34,35]. However, gastrin and the closely related cholecystokinin have been shown to be expressed in the stomach and the central nervous system (CNS) in vertebrates [36], and our analyses here provide further evidence of pleiotropic roles of these genes.

Recent research has found novel roles of gastrin and its related isoforms (progastrin and the glycine-extended gastrins) in cell proliferation, apoptosis and extracellular remodelling in colonic epithelium in a suite of vertebrate models [37]. Furthermore, mouse gastrin knockout models present with heavily deleterious gastric phenotypes: achlorydic gastric environments with high incidences of gastric metaplasia, bacterial overgrowth and tumours [38]. These unique roles and deleterious knockout phenotypes raise questions as to how the monotreme ancestor and potentially other fish lineages could have lost gastrin without a subsequent substantial reduction in fitness.

### 4.2. Evolution of Gastric Genes Implicated in the Secondary Loss of Gastric Phenotype in Jawed Vertebrates (Gnathostomata)

We noted the absence of gastrokine (*Gkn1* and *Gkn2*) and trefoil factor homologs (*Tff1* and *Tff2*), *Vsig1* and *Anxa10* in all agastric taxa with the exception of gastrokine-like sequences and *Vsig1* in the West African lungfish. Convergent loss between agastric Actinopterygii taxa, the West African lungfish and the platypus suggests a role of these genes in the secondary loss of stomach features, in line with the loss of gastric enzymes, proton pump subunits and gastrin [7,9,11,32]. Furthermore, the different inactivating mutations displayed between independent agastric taxa highlight the independent nature of these loss events.

Our findings are not consistent for the echidna, which has an agastric phenotype and has retained *Gkn1*, *Tff2*, *Vsig1* and *Anxa10*. These findings align with our recent works in which the common monotreme ancestor likely lost antral glandular epithelium and pyloric restriction through *Nkx3.2* pseudogenisation and subsequent shifts in developmental processes. However, these findings also point to a retention or re-establishment of pyloric-like restriction in the echidna lineage through potential compensation or an additional evolutionary event [39].

The presence of trefoil factor and gastrokine homologs in Actinopterygii taxa suggests the evolution of these genes likely occurred early in Gnathostome evolution, prior to the divergence of major jawed vertebrate lineages. Furthermore, presence of a homologous trefoil factor-like sequence in the vase tunicate (*Ciona intestinalis*) provides evidence of gene evolution, in line with the advent of the alimentary canal in early ascidians (Appendix A), [3]. Recent research has implicated these genes in the differentiation of squamous and glandular epithelium in the mammalian hindstomach as well as mucosal proliferation/differentiation and protection to chemical insults [40,41,42]. The gastrokines and trefoil factor homologs, in particular, are known for their potent anti-tumorigenic properties and are implicated in several gastrointestinal cancers [43,44,45]). Further in vivo experimentation and comparative genomics in a variety of biological models could determine whether these features represent ancestral traits or lineage-specific innovations.

Moreover, this study highlights that *Anxa10* and *Vsig1* arose prior to the divergence of jawed vertebrates. Previous characterisations of *Anxa10* have outlined its likely role as a calcium-dependent phospholipid-binding protein and its expression in both healthy and cancerous (i.e., carcinoma) gastric tissues in humans, but this study is the first to implicate the gene in function in a wide range of vertebrate species [19,46,47,48]. Similarly, *Vsig1* has been implicated to regulate the development of glandular gastric epithelium in mice models, but our findings, along with those from Kato et al. (2024), show an ancestral role in gastric development in jawed vertebrates [20].

## 5. Conclusions

Here, we investigated the evolution, selection and conservation of the gastrin physiological pathway and other genes involved in gastric function in gastric and agastric jawed vertebrates in order to further understand the independent evolutionary trajectories of the agastric phenotype. Interestingly, we discovered that the absence of a gastric gene repertoire (*Gkn1*, *Gkn2*, *Tff1*, *Tff2*, *Vsig1* and *Anxa10*)—which evolved prior to the divergence of major jawed vertebrate lineages—correlates with the agastric phenotype, with the exception of the echidna and West African lungfish, which may have retained some aspects of gastric functionality (Figure 4). We found that the loss of *Gast* and *Grpr* does not correlate with the gastric or agastric phenotype. These findings differentiate aspects of the agastric phenotype and reveal convergent gene loss and retention in agastric species separated by more than 500 million years of evolution.

## Figures and Tables

**Figure 1 jdb-13-00027-f001:**
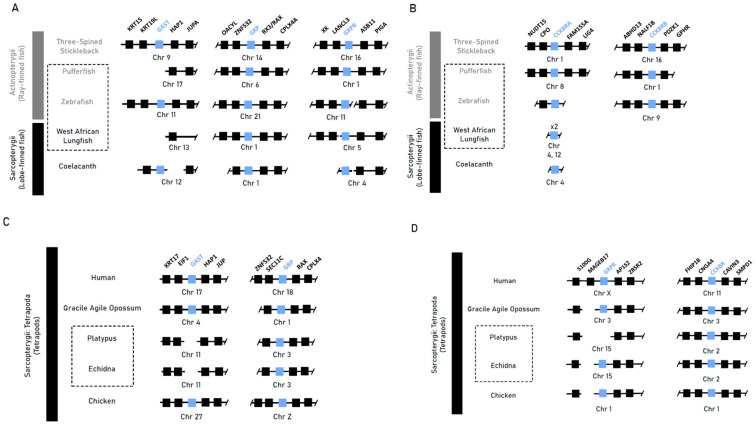
Syntenic analysis of gastrin (*Gast*), gastrin-releasing peptide (*Grp*), gastrin-releasing peptide receptor (*Grpr*) and cholecystokinin receptor B (*Cckbr*) in select vertebrate species. (**A**,**B**) Synteny plot of gastric genes lost or pseudogenised in the agastric species of the zebrafish (*D. rerio*) and West African lungfish (*P. annectens*) compared to the gastric three-spined stickleback (*G. aculeatus*) and coelacanth (*L. chalumnae*). (**C**,**D**) Synteny plot of the short-beaked echidna (*T. aculeatus*) and platypus (*O. anatinus*) when compared to humans (*H. sapiens*/placental mammals), gracile agile opossums (*G. agilis*/marsupials) and chickens (*G. gallus*/reptiles). Class-level taxonomy (Sarcopterygii vs Actinopterygii) is displayed on the left, boxes with dotted lines indicate agastric species, coloured boxes indicate genes of interest, numbers within a box indicate presence of only that family member, and breaks or missing boxes in the syntenic block indicate gene absence.

**Figure 2 jdb-13-00027-f002:**
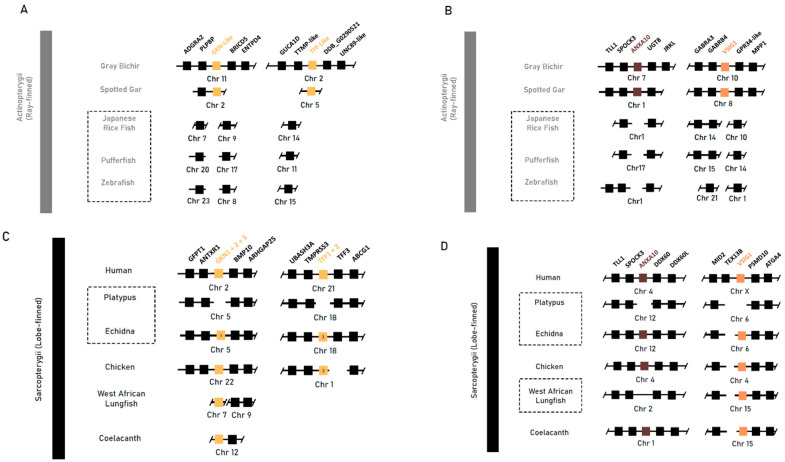
Syntenic analysis of genes implicated in gastric mucosal maintenance (*Gkn1* and *Gkn2*) and gastric development (*Vsig1*) and unclear roles in gastric mucosal maintenance, growth and differentiation (*Anxa10*) in select vertebrate species. (**A**,**B**) Syntenic analyses of Actinopterygii taxa: the agastric teleost species of the Japanese rice fish (*O. latipes*), pufferfish (*T. rubripes*) and zebrafish (*D. rerio*) compared to the gastric grey bichir (*P. senegalus*) and spotted gar (*L. oculatus*). (**C**,**D**) Syntenic analyses of Sarcopterygii taxa: the agastric short-beaked echidna (*T. aculeatus*), platypus (*O. anatinus*) and West African lungfish (*P. annectens*) compared to the gastric human (*H. sapien*), chicken (*G. gallus*) and coelacanth (*L*. *chalumnae*). Class-level taxonomy (Sarcopterygii vs Actinopterygii) is displayed on the left, boxes with dotted lines indicate agastric species, coloured boxes indicate genes of interest, numbers within a box indicate presence of only that family member, and breaks or missing boxes in the syntenic block indicate gene absence.

**Figure 3 jdb-13-00027-f003:**
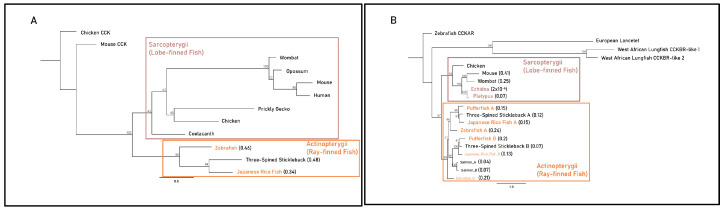
Maximum-likelihood amino acid phylogenies of the gastrin release pathway constituents (**A**) gastrin (GAST), (**B**) cholecystokinin B receptor (CCKBR), (**C**) gastrin-releasing peptide (GRP) and (**D**) gastrin-releasing peptide receptor (GRPR) for select gastric and agastric vertebrate species constructed by the IQ-TREE web server with 1000 bootstrap replicates. Taxa, substitution models as chosen by the Bayesian information criterion (BIC), outgroups and sequence references are in Appendix A. Paralogues of the CCKBR receptor for teleost lineages are identified with the letters A and B. Branch labels denote the bootstrap percentage (%), coloured tip labels represent agastric lineages, coloured boxes outline class-level taxonomic groupings and branch length, and scale bar and bracketed numbers measure substitutions per site to the nearest bifurcation.

**Figure 4 jdb-13-00027-f004:**
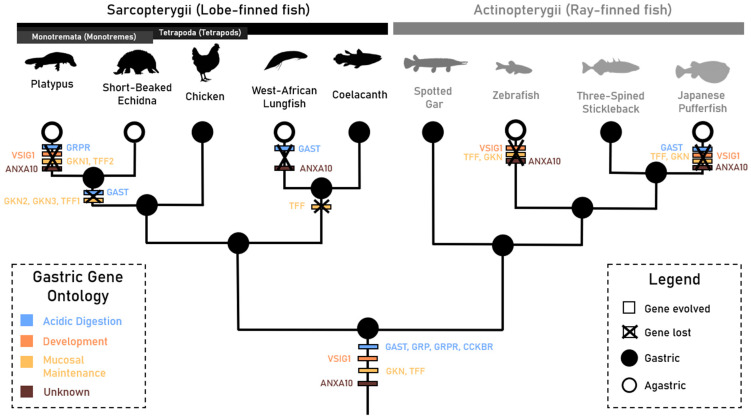
Phylogeny outlining the proposed gastric gene evolution, duplication and loss events between taxa displaying the gastric and agastric phenotype in Sarcopterygii (lobe-finned fish) and Actinopterygii (ray-finned fish) clades. Agastric taxa are denoted by hollow branch tips, gastric gene ontologies are outlined by gene colouration, and gene evolution and loss events are signified by white and black borders, respectively. Branches are not drawn to scale.

**Table 1 jdb-13-00027-t001:** Summary of significant findings (*p* < 0.05) from CodeML branch model selection analysis of *Cckbr*, *Grp* and *Grpr* sequences with columns for the foreground branch (taxonomic grouping or species), the model-fitting metric (likelihood-ratio test *p*) and the Ka/Ks ratio under differing foreground scenarios (foreground = ω1 and background = ω0).

Gene	Foreground Branch	LRT *p* Value	Foreground Ka/Ks (ω1)	Background Ka/Ks (ω0)
*Cckbr*	Japanese Rice Fish B, Pufferfish B and Three-Spined Stickleback B	0.04	0.19	0.08
*Cckbr*	Three-Spined Stickleback B	1.12 × 10^−5^	0.33	0.07
*Cckbr*	Three-Spined Stickleback A	0.01	3 × 10^−3^	0.08
*Grp*	Tetrapoda (Mammals and Reptiles)	2.17 × 10^−5^	8.9 × 10^−4^	0.28
*Grp*	Mammalia	0.03	0.02	0.24
*Grp*	Mouse	0.04	0.06	0.24
*Grpr*	Pufferfish	0.02	0.11	0.04
*Grpr*	Sarcopterygii (Mammals, Reptiles and Lobe-Finned Fish)	8.93 × 10^−7^	1.94 × 10^−3^	0.05

## Data Availability

All sequence data are openly available through NCBI GenBank (https://www.ncbi.nlm.nih.gov/genbank/), otherwise all data necessary to evaluate results and conclusions are available within the paper, references and Appendix A.

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
