# Peer review of "Evolution of the Jawed Vertebrate (Gnathostomata) Stomach Through Gene Repertoire Loss: Findings from Agastric Species"

_jdb, 2025, doi:10.3390/jdb13030027_

Round 1
Reviewer 1 Report
Comments and Suggestions for Authors
Dann and Grützner explore in this manuscript the evolution of the agastric phenotype in the Actinopterygii and Sarcopterygii clades. This represents one of the most striking cases of phenotypic recurrent evolution in multiple vertebrate clades. They add to a set of studies showing the role of gene loss, underlying the repeated emergence of a similar phenotype in different clades. Overall, the work is well performed and solid and I thus recommend publication.
I have one major topic that I think the authors should clarify regarding the timing of origin of Gkn1, Gkn2, Tff1 and Tff2. That is critical to infer loss: “you can only lose the gene that had emerged already”. I would advise some phylogenetics analysis to determine orthology and duplication patterns.
Author Response
We thank the reviewer for their comments on the report and invite them to review the word document which has a detailed response and the corresponding changes.

Reviewer 2 Report
Comments and Suggestions for Authors
This study outlines the evolutionary trajectory of the stomach in vertebrates, starting with the Actinopterygii (many of which lack a stomach), progressing to the Sarcopterygii (which possess a stomach capable of secreting acid and digestive enzymes), and finally to the Monotremata (e.g., the echidna, which do not secrete gastric acid or pepsin). The work focuses on the evolution, selection, and loss of the gastrin physiological pathway, along with genes implicated in gastric development. The findings suggest that the loss of gastrin is unlikely to be associated with the agastric phenotype and a convergent repertoire of gastric genes lost in agastric species, partial retention of gastric functionality in the echidna. Furthermore, this research highlights that Anxa10 and Vsig1 arose prior to the divergence of jawed vertebrates. Below are some suggestions which may help to improve the manuscript.
- The authors’ method for defining gene loss is arbitrary. Gene loss is often influenced by factors such as gene annotation, individual variation, and assembly errors, leading to false-positive results. Please use tools like TOGA or miniprot to confirm whether the discussed genes are indeed missing in the genome. Clarify the type of gene loss (e.g., premature termination, frameshift mutation, or others) and validate the authenticity of these losses using raw sequencing reads.
- The selection analyses feel somewhat disconnected from the preceding sections. The authors found that the loss of Gast and Grpr was not correlated with the stomach-less phenotype, yet proceeded with selection analysis on these genes. Conversely, they identified a set of gastric genes (Gkn1, Gkn2, Tff1, Tff2, Vsig1, Anxa10) whose loss was associated with stomach absence—but did not perform selection analysis on them. Please clarify the rationale for conducting selection analyses in these cases.
- The background part of the abstract is too lengthy; please enhancing the new findings of the article.
- Figures1, Figure2, and Figure3 can be improved:
1) Incorporate a phylogenetic tree with silhouettes of the corresponding species at each branch, indicating whether they are gastric or agastric;
2) In Figures1 and Figure2, highlight key genes such as ANXA10, using colors other than black for better visual distinction;
3) Improve the aesthetic design of the phylogenetic tree in Figure3, referring to examples from the literature (DOI: 10.1126/science.aav6202; DOI: 10.1016/j.cell.2021.01.047) for guidance.
- In this research, the authors primarily focused on genes related to the gastrin physiological pathway and found that gastrin loss is not associated with the agastric phenotype. Aside from the genes (Gkn1, Gkn2, Tff1, Tff2, Vsig1, Anxa10) that are significantly associated with the agastric phenotype, did the authors perform any omics analyses, such as RNA-seq, to first identify a list of genes that are specifically and highly expressed in gastric tissue, followed by GO and KEGG pathway enrichment analysis?
Author Response
We thank the reviewer for their comments and suggestions and invite them to review the attached document outlining these changes and how they have been incorporated in the manuscript.

Reviewer 3 Report
Comments and Suggestions for Authors
This study addresses the phenomenon of “stomach loss” that seems to have occurred independently in several vertebrate lineages, including monotremes, lungfish, and teleosts. This condition is marked by the absence of gastric glands, acid secretion, and storage capacity. While key gastric genes such as Pga, Pgc, and Atp4 subunits are consistently lost in agastric species, the role of Gast (gastrin) and its pathway has not been fully explored. This study found no clear correlation between the loss of Gast or Grpr and the agastric phenotype. However, a set of gastric genes (e.g., Gkn1, Tff2, Vsig1) was lost in all agastric species except the echidna, which retained some. Thus, results support convergent loss of gastric genes across agastric lineages and suggest that gastrin loss alone does not define the agastric phenotype.
It is an important study for understanding the evolution of gastric function, particularly by providing new insights into the conservation of a novel gastric gene repertoire (Gkn1, Gkn2, Tff1, Tff2, Vsig1, Anxa10) and genes related to the gastrin pathway.
However, for certain claims made by the authors, it would have been ideal to include analyses using the elephant shark genome, as it represents a lineage that predates the divergence of Actinopterygii and Sarcopterygii.
In addition, the text could be further improved, and I noticed that some references are not appropriate. I recommend that the authors carefully review all cited references. Please find specific comments below.
- Abstract:
In my opinion, the last sentences referring to the main results can be improved.
- Introduction
- In the first paragraph of the introduction ( Pag. 1, Line 30), I would suggest including an additional reference to a study highlighting the strong conservation of molecular mechanisms in sharks, which are considered representative of basal gnathostomes. To incorporate this idea, I have added a segment to the sentence (underlined and in bold below).
Sentence changed:
“The stomach is a highly conserved and integral organ throughout jawed vertebrate 30 (Gnathostomata) evolution as evidenced by conserved developmental pathways found in extant representatives of basal Gnathostomes (Gonçalves et al., 2019) and even dating back to protochordate species (Nakazawa et al., 2013; Nakayama et al., 2017)”.
Reference:
Molecular ontogeny of the stomach in the catshark Scyliorhinus canicula. Gonçalves O, Freitas R, Ferreira P, Araújo M, Zhang G, Mazan S, Cohn MJ, Castro LFC, Wilson JM. Sci Rep. 2019 Jan 24;9(1):586. doi: 10.1038/s41598-018-36413-0.
- In the first sentence on 2, I would suggest some edits to enhance clarity, as indicated below (underlined and in bold below).
Sentence changed:
“In line with convergent phenotypes, agastric genotypes also exhibit similarities: agastric species share a concomitant loss of genes encoding gastric enzymes (Pga, Pgc) and proton pump subunits (Atp4a, Atp4b).
- The second sentence on 2 also requires corrections (underlined and in bold below).
Sentence changed:
“Previously, it was found that monotremes have also lost gastrin (Gast), a key gastric hormone regulated by the upstream neuropeptide gastrin-releasing peptide (Grp) and its native receptor (Grpr). When released from gastric G cells, gastrin binds to the cholecystokinin B receptor (Cckbr) on parietal cells, thereby potentiating gastric acid secretion.
- The sentence on 2 line 67 can be improved and divided in two (underlined and in bold below).
Sentence changed:
“Despite extensive work on expression patterns (particularly in the gastrokine and trefoil factor families) and a recent study by Kato et al. (2024), which reported the absence of Vsig1 in agastric teleost lineages and the platypus, important questions remain. In particular, it is still unclear what emergent properties (e.g., gene ontologies) arise from the set of lost genes, and whether these additional genes are conserved across agastric vertebrate lineages (Jiang, Lossie & Applegate, 2011; Geahlen et al., 2013).”
- Results
- The last sentence of the results on Pag. 3 should be changed (underlined and in bold below).
Sentence changed:
“These findings suggest that the agastric phenotype does not correlate with the loss of gastrin. Furthermore, the absence of gastrin is not consistently associated with the loss of its indirect upstream regulator, Grpr.
- The last sentence of the results on Pag. 4 should be changed (underlined and in bold below).
Sentence changed:
“These findings suggest that these genes were present in Vertebrate genomes prior to the evolutionary split between Actinopterygii and Sarcopterygii, and were later lost in particular vertebrate lineages.”
Additional analyses:
The authors have an opportunity to explore this hypothesis by analysing these genes in the elephant shark, which—given its phylogenetic position—serves as a representative of basal gnathostomes, predating the evolutionary split between Actinopterygii and Sarcopterygii. There is also a previous work that shows that a particular species of shark has gastrin (Proc Natl Acad Sci U S A. 1997 Sep 16;94(19):10221–10226. doi: 10.1073/pnas.94.19.10221) and the authors should comment on that and search for more “shark” references concerning these genes.
- The last sentence of the results on Pag. 6 should be changed (underlined and in bold below).
Sentence changed:
“Together, these findings suggest that the presence or absence of a stomach does not strongly correlate with accelerated molecular evolution or pseudogenization of these genes across vertebrate lineages. Instead, some agastric species retain high sequence conservation, implying possible pleiotropic roles or constraints unrelated to gastric function.”
- The last sentence of the results on Pag. 7 should be changed (underlined and in bold below).
Sentence changed:
“These findings suggest that independent bursts of sequence evolution in the gastrin release pathway were primarily characterised by conservative mutations resulting in the maintenance of functional integrity. Furthermore, these molecular changes showed no direct correlation with variation in gastric phenotypes.”
Discussion
- Discussion
- 8 line 243: delete “surprisingly”
- 8 line 254: I do not entirely agree with the statement “ However, it remains unclear as to why agastric species predominantly lost Gast and why the platypus and three-spined stickleback would lose Grpr…..”
My opinion is that the fact that different agastric lineages have lost different components of the gastrin signalling pathway, such as Gast or Grpr, is consistent with the idea that the loss of gastric function occurred independently across vertebrate clades, rather than through a shared evolutionary mechanism.
- 9 line 282: I do not entirely agree with the statement “The presence of trefoil factor and gastrokine homologs in Actinopterygii taxa suggests the evolution of these genes predates the divergence of jawed vertebrates (Gnathostomata).” I also do not agree with the references used to support this statement: 1) it is not Jiang & Applegate, but Jiang, Lossie & Applegate and the article is in Gallus Gallus Domesticus, wich is not an Actinopterygii; 2) Geahlen et al is a study on human gastrokine locus and confounding factor, again… not Actinopterygii.
My opinion is that the presence of trefoil factor and gastrokine homologs in Actinopterygii cannot suggest that these genes evolved before the emergence of jawed vertebrates (Gnathostomata), since Actinopterygii are themselves gnathostomes. A more accurate interpretation is that these genes likely originated early in gnathostome evolution, prior to the divergence of major jawed vertebrate lineages. Again, analyses on sharks are crucial to better understand the ancestry of these genes.
- 9 line 282: I do not entirely agree with the statement “Further in vivo experimentation in a variety of biological models could determine whether these are newly-acquired functions or whether these features date back to early jawed-vertebrates (Oidovsambuu et al., 2011, Gerke et al., 2005; Tsai et al., 2015).” I also do not understand the use of these references here.
My opinion is that further in vivo experimentation in a variety of biological models could help elucidate the functional roles of these genes, but additional comparative genomic and phylogenetic analyses are also crucial to clarify whether these features represent ancestral traits or lineage-specific innovations. Again, studies on sharks will help.
- 9 line 282:” expression in gastric and gastric carcinoma tissues in humans”. This should be clarified: gastric and gastric carcinomas?
- 9 line 298: I do not agree with “ this study is the first to implicate the gene in gastric development and function in a wide range of vertebrate species”. In the manuscript, there is no clear reference to “gastric development”
The English is generally good, but some sentences could be improved for greater clarity. I have provided specific comments to the authors regarding these.
Author Response
We thank the reviewer for their comments and suggestions and invite them to review the attached document which details these and how they have been incorporated in the manuscript.

Round 2
Reviewer 2 Report
Comments and Suggestions for Authors
The authors have addressed most of my concerns.